# Performance of Scoring Systems in Predicting Clinical Outcomes in Patients with Bacteremia of *Listeria monocytogenes*: A 9-Year Hospital-Based Study

**DOI:** 10.3390/biology10111073

**Published:** 2021-10-21

**Authors:** Shang-Hsuan Huang, Ming-Shun Hsieh, Sung-Yuan Hu, Shih-Che Huang, Che-An Tsai, Chiann-Yi Hsu, Tzu-Chieh Lin, Yi-Chen Lee, Shu-Hui Liao

**Affiliations:** 1Department of Emergency Medicine, Taichung Veterans General Hospital, Taichung 40705, Taiwan; empor20267@gmail.com (S.-H.H.); h852@vghtc.gov.tw (T.-C.L.); 2Department of Emergency Medicine, Taichung Armed Forces General Hospital, Taichung 40466, Taiwan; 3Department of Emergency Medicine, Taipei Veterans General Hospital, Taoyuan Branch, Taoyuan 330, Taiwan; edmingshun@gmail.com (M.-S.H.); leeyichen9@yahoo.com.tw (Y.-C.L.); 4Department of Emergency Medicine, Taipei Veterans General Hospital, Taipei 11217, Taiwan; 5School of Medicine, National Yang Ming Chiao Tung University, Taipei 11221, Taiwan; 6School of Medicine, National Chung Hsing University, Taichung 402, Taiwan; 7School of Medicine, Chung Shan Medical University, Taichung 40201, Taiwan; cucu0214@gmail.com; 8Institute of Medicine, Chung Shan Medical University, Taichung 40201, Taiwan; 9Department of Nursing, College of Health, National Taichung University of Science and Technology, Taichung 404, Taiwan; 10Department of Emergency Medicine, Chung Shan Medical University Hospital, Taichung 40201, Taiwan; 11Lung Cancer Research Center, Chung Shan Medical University Hospital, Taichung 40201, Taiwan; 12Division of Infectious Disease, Department of Internal Medicine, Taichung Veterans General Hospital, Taichung 40705, Taiwan; lucky-sam@yahoo.com.tw; 13Biostatistics Task Force, Department of Medical Research, Taichung Veterans General Hospital, Taichung 40705, Taiwan; chiann@vghtc.gov.tw; 14Department of Pathology and Laboratory, Taipei Veterans General Hospital, Taoyuan Branch, Taoyuan 330, Taiwan; selina620626@gmail.com

**Keywords:** area under the curve (AUC), bacteremia, *Listeria monocytogenes* (LM), mortality in emergency department sepsis (MEDS) score, national early warning score (NEWS), scoring systems, receiver operating characteristic curve (ROC)

## Abstract

**Simple Summary:**

Listeria monocytogenes (LM) may develop life-threatening invasive infections with a mortality rate of 25–30%. The aim of this study is to investigate the scores of LM bacteremia to predict the clinical outcomes. A total of 38 patients were studied, including 16 males (42.1%) and 22 females (57.9%), with a mean age of 59.9 ± 19.6 years. The hospital stay averaged 23.3 ± 20.9 days with an in-hospital mortality rate of 36.8%. The Mortality in Emergency Department Sepsis (MEDS) Score was 6.6 ± 4.0 for survivors and 12.4 ± 4.4 for non-survivors (P < 0.001). The National Early Warning Score (NEWS) was 3.9 ± 2.8 for survivors and 7.8 ± 3.1 for non-survivors (P = 0.001). The AUC of ROC was 0.829 for MEDS and 0.815 for NEWS in predicting the mortality risk. MEDS (≥10) and NEWS (≥8) were both good predictors of the clinical outcome in LM bacteremia. Further large-scale studies are required to gain a deeper understanding of this disease and to ensure patient safety.

**Abstract:**

Background: *Listeria monocytogenes* (LM) is a facultative anaerobe, Gram-positive bacillus which is widely distributed in nature, and can be separated from soil, water, and rotten vegetables. Immunocompetent people are less likely to suffer from LM infection or may only show gastrointestinal symptoms. However, immunocompromised elderly people, pregnant women, and newborns may develop life-threatening invasive infections. The mortality rate of LM infection is as high as 25–30%. The aim of this study is to investigate clinical scores of patients with bacteremia of LM confirmed by one or more blood cultures. We analyzed their demographics and laboratory findings in relation to their clinical outcomes. Materials and Methods: This was a hospital-based retrospective study on patients with bacteremia of LM. Data were collected from the electronic clinical database of Taichung Veterans General Hospital between January 2012 and December 2020. Bacteremia of LM was confirmed by at least one blood culture. Demographics, clinical characteristics, and laboratory data were collected for analysis. A variety of clinical scoring systems were used to predict the clinical outcome. Results: A total of 39 patients had confirmed bacteremia of LM. Among them, 1 neonatal patient was excluded. The remaining 38 patients were studied. They included 16 males (42.1%) and 22 females (57.9%), with a mean age of 59.9 ± 19.6 years. Their hospital stay averaged 23.3 ± 20.9 days. The in-hospital mortality rate was 36.8%. Mortality in Emergency Department Sepsis (MEDS) Score was 6.6 ± 4.0 for survivors and 12.4 ± 4.4 for non-survivors (P < 0.001). The National Early Warning Score (NEWS) was 3.9 ± 2.8 for survivors and 7.8 ± 3.1 for non-survivors (P = 0.001). Regarding the prediction of mortality risk, the AUC of ROC was 0.829 for MEDS and 0.815 for NEWS. Conclusions: MEDS and NEWS were both good predictors of the clinical outcome in LM bacteremic patients. In those with higher scores of MEDS (≥10) and NEWS (≥8), we recommended an early goal-directed therapy and appropriate antibiotic treatment as early as possible to reduce mortality. Further large-scale studies are required to gain a deeper understanding of this disease and to ensure patient safety.

## 1. Introduction

Listeriosis is a disease typically caused by the infection of *Listeria monocytogenes* (LM). It is mostly transmitted through contaminated food [1,2]. LM is a Gram-positive facultative intracellular bacterium, which is widely distributed in nature, and it can be cultivated from soil, water, milk, and rotten vegetables [3,4,5]. *Listeria* can also be transmitted from a pregnant woman to her fetus via the placenta or exposure during delivery [6]. Immunocompetent people are less likely to suffer from LM infection or may only show gastrointestinal symptoms such as diarrhea, nausea, and vomiting [5]. However, in the elderly, immunocompromised people, pregnant women, and newborns, LM may develop invasive infections such as sepsis, infecting the central nervous system (CNS) with life-threatening consequences [7,8].

According to the World Health Organization, the mortality rate of LM is around 20–30% [9]. The Taiwan Centers for Disease Control reported 168 local cases of Listeriosis in 2018, corresponding to an incidence rate of 0.72 case/100,000 people, with a 25% mortality rate [10]. Risk factors for mortality are type 2 diabetes mellitus (DM), old age, steroid use, respiratory distress, solid organ malignancy, and hepatic decompensation [11,12,13]. In the western world, risk factors for similar mortality are alcoholism, kidney disease, cardiovascular disease, immunosuppression, septicemia, multi-organ failure, and monocytopenia [14,15,16].

Bacteremia is a life-threatening critical condition [17]. Some predictive scoring models have been established on the mortality from bacteremia. For instance, the Pitt bacteremia score (PBS) predicts mortality in patients with bloodstream infections and has been in use for 3 decades to stratify the severity of illness [18]. The Gram-negative bloodstream infection (BSI) risk score predicts the 28-day mortality of Gram-negative BSI [19]. In addition, many scoring systems have been developed to predict the mortality risk in the emergency department (ED). Their effectiveness has been reported under different situations, such as infectious disease, length of stay, and hospital admission [20,21]. In a recent study, the Rapid Emergency Medicine Score (REMS) was applied to patients with COVID-19 and shown to be effective in its risk stratification [22].

Reviewing the literature, we found no specific scoring systems to predict the mortality of *Listeria* bacteremia. Due to its low incidence but high mortality rate, we here aimed to validate the performance of a variety of clinical scoring systems (*n* = 6) to assess the severity and clinical outcomes of this disease. We investigated the epidemiology and clinical characteristics of LM bacteremia and applied these clinical scoring systems to predict the risk of mortality.

## 2. Materials and Methods

### 2.1. Data Collection and Definition

Our study was approved by the institutional review board of Taichung Veterans General Hospital (No. CE21215A). It was a hospital-based retrospective study on patients with LM bacteremia. Cases of confirmed LM bacteremia were each based on the results from at least one blood culture in the ED. Patient data were extracted from the electronic clinical database of Taichung Veterans General Hospital, covering a period from January 2012 to December 2020. Data were demographics, laboratory investigations, and clinical outcomes. The in-hospital mortality was the primary outcome. Univariate and multivariate analyses were used to evaluate the mortality risk. Immunocompromised conditions included neoplasm, chronic kidney disease, chronic liver disease, recipient of transplant, and autoimmune disease.

### 2.2. Scoring Systems

For clinical outcome and mortality risk, the clinical scoring systems we had analyzed were as follows: Mortality in Emergency Department Sepsis (MEDS) Score, Modified Early Warning Score (MEWS), National Early Warning Score (NEWS), Rapid Acute Physiology Score (RAPS), Rapid Emergency Medicine Score (REMS), and quick Sequential Organ Failure Assessment (qSOFA).

### 2.3. Statistical Analyses

Continuous data were expressed as mean ± standard deviation (SD). Categorical data were expressed as number and percentage. Chi-squared tests were used to compare categorical data, and Mann–Whitney–Wilcoxon U tests used to compare continuous data, regarding mortality risks in survivors and non-survivors. Univariate and multivariate analyses were performed using the Cox regression model to assess possible predictors for mortality, and results expressed as hazard ratio and confidence interval. We used the area under the curve (AUC) of receiver operating characteristic curve (ROC) to compare predictive power across different scoring systems. Cut-off points were used to stratify mortality risks in terms of sensitivity, specificity, negative predictive value (NPV), and positive predictive value (PPV). P values < 0.05 were considered statistically significant. Analyses were performed on the Statistical Package for the Social Science (IBM SPSS version 22.0; International Business Machines Corp, New York, NY, USA).

## 3. Results

### 3.1. Demographics and Clinical Characteristics

A total of 39 patients had confirmed LM bacteremia. Among them, 1 neonatal patient was excluded. Demographic and clinical findings of the remaining 38 patients are summarized in Table 1. A total of 38 patients enrolled in the study, including 16 males (42.1%) and 22 females (57.9%), with their mean age at 59.9 ± 19.6 years. The annual incidence rate was 0.52 cases per 10,000 ED visits. The average length of hospital stay was 23.3 ± 20.9 days. The associations between years distribution with survival and mortality were shown in Figure 1 (P = 0.264). Seasonal variations between average air temperature and patient distribution in each season were shown in Figure 2 (P = 0.226).

### 3.2. Microbiology

Bacterial culture from individual patients was performed at least once. Seven patients had concomitant positive results of cultures from blood and from cerebrospinal fluid. Only one patient had positive LM results of cultures detected in both blood and ascites.

### 3.3. Clinical Syndromes and Management

Major clinical syndromes were divided into five categories: fever, gastrointestinal (GI) symptoms, respiratory symptoms, neurological symptoms, and others. Fever was the leading symptom (68.4%), followed by GI symptoms (34.2%) and neurological symptoms (31.6%). There were more non-survivors using oxygen (33.3% vs. 100.0%, P < 0.001) and vasopressors (4.2% vs. 35.7%, P = 0.0018). Their average body temperature was 38.1 ± 1 °C, with no difference between the survivor and non-survivor groups (Table 1).

### 3.4. Laboratory Data and Scoring Systems

Laboratory data and scoring systems are summarized in Table 1. Levels of c-reactive protein (CRP) (8.3 ± 7.8 vs. 14.3 ± 5.4, P = 0.003) and lactic dehydrogenase (LDH) (306.9 ± 110.0 vs. 770.3 ± 706.0, P = 0.016) were significantly higher in non-survivors. The non-survivors had significantly higher scores in the scoring systems of MEDS (6.6 ± 4.0 vs. 12.4 ± 4.4, P < 0.001), NEWS (3.9 ± 2.8 vs. 8.0 ± 3.0, P = 0.001), and qSOFA (0.4 ± 0.5 vs. 0.9 ± 0.8, P = 0.041).

### 3.5. Primary Outcomes and Comorbidities

Fourteen patients died, equivalent to a mortality rate of 36.8%. Eleven patients had bacteremia and meningitis of LM together. In the neoplastic group, solid organ tumors (*n* = 10, 26.3%) and hematologic disease (*n* = 9, 23.7%) constituted half of these patients. In the group with solid organ tumors, lung cancer (*n* = 3) was the most common type, followed by colon cancer (*n* = 2) (Table 1).

### 3.6. Univariate and Multivariate Analysis of Risk Factors

Univariate analyses for predisposing factors were conducted on clinical outcomes in these patients, with results summarized in Table 2. We found higher hazard ratios (HR) in non-survivors for the following: respiratory syndromes, shock, respiratory rate, neoplasm, elevated liver enzymes (alkaline phosphatase [ALK-P], alanine aminotransferase [ALT], and aspartate aminotransferase [AST]), LDH, CRP, and the usage of vasopressors. Scores of MEWS, MEDS, NEWS, and qSOFA were significantly higher in non-survivors. Multivariate logistic regression analyses for predisposing factors were performed to evaluate the clinical outcomes in these patients, with results summarized in Table 3. We found higher HR in non-survivors regarding scores of both MEDS (P = 0.001) and NEWS (P = 0.003).

### 3.7. Receiver Operating Characteristic Curve (ROC)

We analyzed ROC of both MEDS and NEWS for the accuracy in predicting mortality risks, and results shown in Figure 3 and Table 4. The cut-off point of MEDS was 10, and the area under the curve (AUC) of ROC measured up to 0.829 had a sensitivity of 78.6% and a specificity of 79.2% (P = 0.001). The cut-off point of NEWS was 8, and the AUC of ROC reached up to 0.815 had a sensitivity of 57.1% and a specificity of 91.7% (P = 0.001).

## 4. Discussion

The reported mortality rate of Listeriosis was about 25~30% [12,15]. Our present study showed a higher mortality rate of 36.8%. In addition to mortality, we found a higher incidence rate (11/38, 28.9%) of meningitis in patients of LM bacteremia. *Listeria* is a well-known microorganism of bacterial meningitis, particularly for those immunocompromised. It is the third most common pathogen of bacterial meningitis in adults [23,24]. The incidence rate of CNS involvement with Listerosis is 22.4% (304/1357) in England and 16.5% (15/115) in Taiwan [12,15]. The prevalence rate of meningitis in LM bacteremia is 7.3% (3/41) in Taiwan and 15.1% (11/73) in Madrid [11,25]. The relationship between CNS infection and mortality remains undetermined. Several studies reported higher mortality rates with CNS involvement, while others reported the opposite [7,14,26]. In the MONALISA cohort study, 3-month mortality with bacteremia was higher than those with neurolisteriosis. Whereas a higher mortality rate was found with neurolisteriosis, especially for those complicated with LM bacteremia [16]. Our study showed a higher rate (28.9%) of meningitis in patients of LM bacteremia with a higher mortality rate (35.7%) compared with previous reports [11,12,15,16]. The higher rates of CNS infection and mortality in our study might be related to our small sample size, and all our cases were LM bacteremia.

Risk factors for mortality in Listeriosis are type 2 DM, old age, steroid use, respiratory distress, solid organ malignancy, hepatic decompensation, alcoholism, kidney disease, cardiovascular disease, immunosuppression, septicemia, multi-organ failure, and monocytopenia [11,12,13,14,15,16]. We have focused on risk factors of mortality related to LM bacteremia. Taiwanese patients with LM bacteremia, when occurring in the elderly, had respiratory distress, and type 2 DM, with a higher mortality rate [11]. According to a Spanish study in Madrid, treatment with non-steroidal immunosuppressive medication and anti-neoplastic therapy are also related to a higher mortality rate [25]. In our univariate analyses, we found risk factors of mortality from LM bacteremia were the following: respiratory syndromes, shock, neoplasm, elevated liver enzymes, LDH, and CRP; higher scores of MED, MEWS, NEWS, and qSOFA; and the usage of vasopressors, particularly patients with neoplasm. These risk factors are consistent with previous studies [11,12,15]. Half of our patients had neoplasms, including 10 (26.3%) with solid organ tumors and 9 (23.7%) with hematologic disorders. In the group with solid organ tumors, lung cancer accounted for the majority (*n* = 3), a proportion that is similar to previous studies [12,25]. Whether patients with lung cancers are more likely to develop LM bacteremia remains to be further investigated.

In general, the clinical presentations were increased demand oxygen, hypotension with hypoxia of tissue, acute kidney and liver injury, leukocytosis, high level of CRP, and even shock in patients with infectious status that progressed to severe disease. In our study, respiratory syndrome and shock, including increased respiratory rate, decreased saturation, elevated level of LDH and CRP, and use of vasopressors, were associated with a higher mortality rate (Table 2). Patients of LM bacteremia with respiratory syndrome tended to have a higher mortality rate, consistent with previous reports [11,12]. However, the relationship and mechanism between respiratory syndrome and mortality of LM bacteremia will be further investigated. Whereas abnormal liver function tests were associated with a higher mortality rate, we speculated that patients in an infectious state may result in hypoperfusion of the liver, thus it was overweighted to be a risk factor for the evaluation of the mortality.

LM is commonly found in nature and in contaminated or uncooked foods [27]. Dairy products such as soft cheese have high risks of LM infection [28]. Many kinds of food have been examined for LM contamination in Taiwan [29], not to ignore the possibility of cross-contamination of foods stored in the freezer [30]. Thirteen of our patients (34.2%) had GI symptoms and only two cases were documented to be food-related (one with raw food, and one with dairy products). Since our data were collected retrospectively, the source of LM bacteremia was not confirmed due to missing information in the chart record on infection source. LM bacteremia is most likely to be related to cross-contamination of foods according to a previous report [30]. Further investigation of the food-borne LM infection is needed.

LM could survive in a variety of environmental conditions over a large temperature range from 1 to 45 °C. Bacteria in food refrigerated at 4 °C can grow within 8–10 days [4,31,32,33,34]. Although our study showed no significant seasonal differences, the outbreak of LM infections tended to occur more often in the summer [35]. In Taiwan, our Spring-like climate year-round with minimal variations could explain the lack of any seasonal change in LM infection.

A number of clinical scoring systems are available to quickly stratify patients and to identify potentially critical conditions in ED and intensive care units based on variable physiological parameters [20,36]. With these simple and easy-to-use clinical scoring systems, physicians can quickly decide on the patient’s treatment options and can start early goal-directed therapy such as appropriate antibiotics treatment. The MEDS score is widely used to predict mortality for patients with community-acquired bacteremia in Taiwan [37]. NEWS is used in America to determine the risk of mortality for inpatients with and without infections [38]. In Thailand, REMS has a higher accuracy in predicting the in-hospital mortality, when compared with sepsis-related scores (qSOFA and systemic inflammatory response syndrome) for patients with suspected sepsis in ED [36].

In this single-center retrospective study, we found higher scores of MEWS, MEDS, NEWS, and qSOFA in non-survivors with the univariate analysis. Furthermore, multivariate logistic regression demonstrated that the AUC of ROC of both MEDS (0.829) and NEWS (0.815) had excellent performance in predicting mortality of LM bacteremia with cut-off points at 10 and 8, respectively. The MEDS score was first developed by Shapiro et al. in 2003. It is based on clinical parameters, including terminal disease, respiratory difficulty, septic shock, platelet count, band proportion, age, lower respiratory infection, nursing home residence, and altered mental status [39]. This score accurately predicts mortality in ED patients with suspected infections [40]. The AUC of ROC for MEDS could measure up to 0.829 with a sensitivity of 78.6% and a specificity of 79.2%. Findings are in support of the excellent discrimination of MEDS in predicting mortality of LM bacteremia. The NEWS score, first developed by the Royal College of Physicians in 2012, consists of respiratory rate, oxygen saturation, temperature, systolic blood pressure, pulse rate, and level of consciousness [41]. This score is more effective than SIRS and qSOFA in early recognition of severe sepsis and septic shock [42]. The AUC of ROC for NEWS could reach up to 0.815 with a sensitivity of 57.1% and a specificity of 91.7%. It is also effective in predicting mortality, but at a sensitivity lower than the MEDS.

Terminal disease is an item in MEDS. In our present study, neoplasm proved to be a risk factor for LM bacteremia. Since MEDS has more scoring items compared with NEWS, it naturally has a higher predictive power in discrimination. For this reason, the AUC of ROC appeared larger for MEDS than for NEWS. MEDS also had a higher sensitivity. Although these two clinical scoring systems are different, we believe that through different cutoff values, they can both accurately predict mortality, facilitating an early initiation of aggressive therapies.

## 5. Limitations

First, this was a single-center study with a small sample size and retrospective in nature. Data might not represent the full characteristics of LM bacteremia. Second, we did not collect prescriptions of antibiotics to evaluate their impact on mortality. Third, clinical symptoms were investigated retrospectively without uniform criteria, resulting in inevitable bias. Fourth, the rarity of patients with LM bacteremia did not favor a prospective randomized control study. All our patients had confirmed diagnosis of LM bacteremia. We did not have those data of initial *Listeria* infection without bacteremia, but that later progressed to severe disease. Prognostic indicators and scoring systems may be designed for LM bacteremia through multicenter study to predict its clinical outcomes based on risk factors.

## 6. Conclusions

Physicians should have a high level of suspicion in bacteremic patients with LM. MEDS and NEWS scores had good performance as prognostic factors to predict clinical outcomes in LM bacteremic patients. Although LDH, CRP, uses of oxygen and vasopressors, and qSOFA, were also prognostic factors for non-survivors of LM bacteremia, in those with higher scores of MEDS (≥10) and NEWS (≥8), we recommended an early goal-directed therapy and appropriate antibiotics as early as possible to reduce mortality. Large-scale studies are required to help further understand this disease and to ensure patient safety.

## Figures and Tables

**Figure 1 biology-10-01073-f001:**
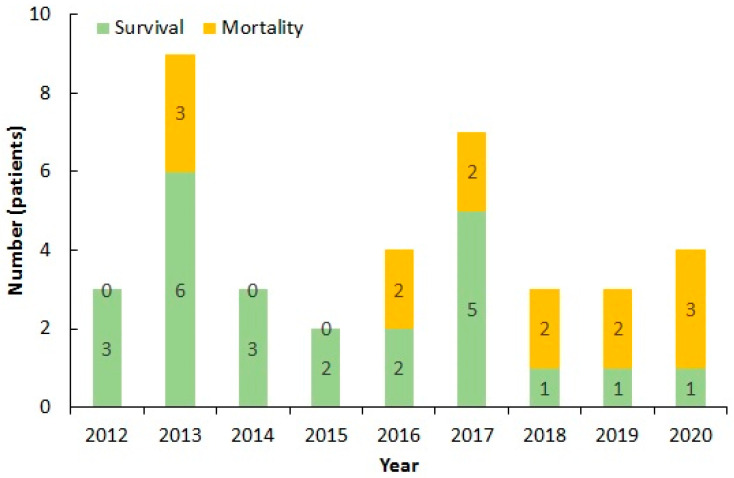
The associations between years distribution with survival and mortality of the 38 patients with *Listeria monocytogenes* bacteremia (P = 0.264). (Green: survival patient number; Yellow: mortality patient number) from 2012 to 2020.

**Figure 2 biology-10-01073-f002:**
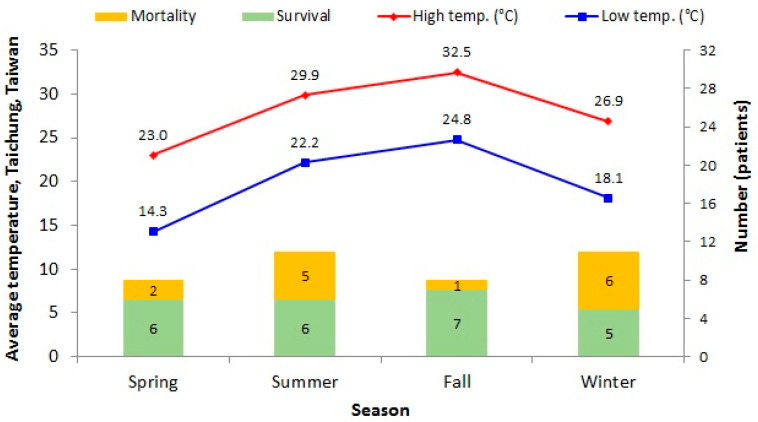
Seasonal variations between average air temperature and patient distribution in each season (P = 0.226). (Green: survival patient numbers; Yellow: mortality patient numbers; Red: average high temperature; Blue: average low temperature).

**Figure 3 biology-10-01073-f003:**
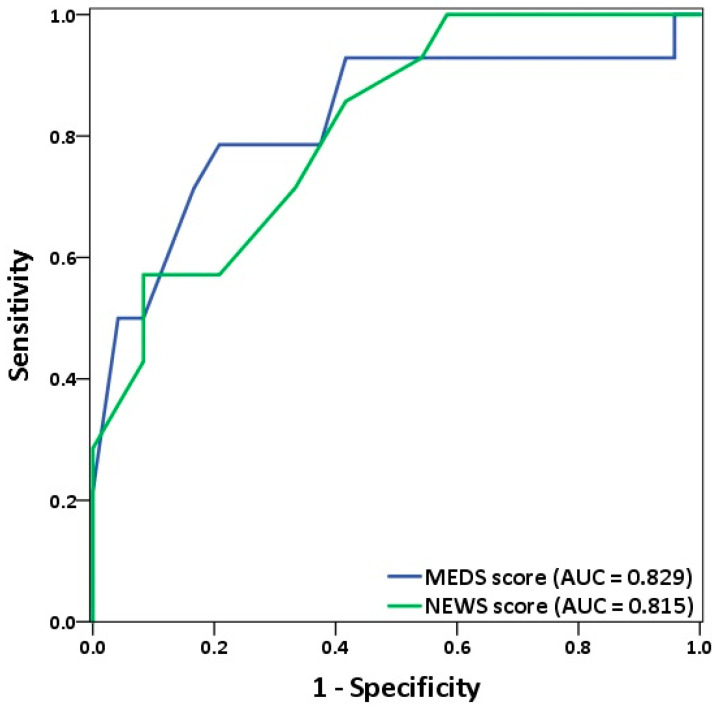
ROC of both MEDS and NEWS analyzed to show the accuracy in predicting mortality risks of LM bacteremia. The AUC of ROC for MEDS and NEWS indicated 0.829 and 0.815, respectively. AUC = Area under the curve; ROC = Receiver operating characteristic curve.

**Table 1 biology-10-01073-t001:** Demographics and laboratory data of 38 patients with *Listeria monocytogenes* bacteremia.

General Data	Patients (*n* = 38)	Survival (*n* = 24)	Mortality (*n* = 14)	P Value
Age (years)	59.9 ± 19.6	57.3 ± 21.4	63.4 ± 15.9	0.732
Male (%)	16(42.1%)	10(41.7%)	6(42.9%)	1.000
LOS (days)	23.3 ± 20.9	25.3 ± 20.5	19.9 ± 21.9	0.232
Clinical syndromes				
Fever	26(68.4%)	18(75%)	8(57.1%)	0.296
GI	13(34.2%)	8(33.3%)	5(35.7%)	1.000
Respiratory	6(15.8%)	2(8.3%)	4(28.6%)	0.167
Neurology	12(31.6%)	7(29.2%)	5(35.7%)	0.728
Other ^a^	1(2.6%)	1(4.2%)	0(0.00%)	1.000
Clinical conditions				
Shock	7(18.4%)	2(8.3%)	5(35.7%)	0.077
Meningitis	11(28.9%)	6(25.0%)	5(35.7%)	0.712
Comorbidities				
Neoplasm	19(50%)	9(37.5%)	10(71.4%)	0.093
Cardiovascular disease	16(42.1%)	9(37.5%)	7(50.0%)	0.680
CKD	10(26.3%)	5(20.8%)	5(35.7%)	0.449
DM	6(15.8%)	4(16.7%)	2(14.3%)	1.000
Chronic liver disease	3(7.9%)	1(4.2%)	2(14.3%)	0.542
Autoimmune	6(15.8%)	6(25.0%)	0(0.00%)	0.067
Immunocompromised	32(84.2%)	19(79.2%)	13(92.9%)	0.383
Maternal–fetal	2(8.33%)	2(8.33%)	0(0.00%)	0.522
Vital signs				
SBP (mmHg)	138.5 ± 28.3	142.5 ± 30.0	131.5 ± 24.6	0.535
MAP (mmHg)	99.7 ± 19.8	104.2 ± 21.0	92.1 ± 15.4	0.209
HR (bpm)	104.3 ± 27.4	102.2 ± 38.2	107.9 ± 26.8	0.606
RR (bpm)	19.9 ± 2.9	19.4 ± 2.2	20.7 ± 3.7	0.381
BT (°C)	38.1 ± 1.0	38.0± 0.8	38.4 ± 1.2	0.525
SpO2 (%)	95.2 ± 5.2	96.0 ± 5.0	93.7 ± 5.4	0.111
GCS	13.7 ± 2.9	14.3 ± 1.6	12.5 ± 4.0	0.209
Laboratory data				
WBC (counts/uL)	10,526.3 ± 6646.4	11,468.3 ± 7034.8	8911.4 ± 5807.4	0.377
Hb (g/dL)	11.0 ± 2.4	11.2 ± 2.5	10.9 ± 2.2	0.397
Plt (×10^3^ counts/uL)	144.0 ± 85.0	160.4 ± 94.7	116.0 ± 56.7	0.215
Cre (mg/dL)	2.1 ± 2.6	2.0 ± 2.7	2.2 ± 2.5	0.341
ALK-P (U/L)	238.4 ± 0.6	158.0 ± 132.8	443.8 ± 591.1	0.054
AST (U/L)	109.3 ± 212.4	41.0 ± 12.8	221.0 ± 322.5	0.07
ALT (U/L)	65.0 ± 63.9	49.0 ± 30.5	91.4 ± 92.2	0.491
LDH (U/L)	529.4 ± 538.9	306.9 ± 110.0	770.3 ± 706.0	**0.016 ***
Lactate (mg/dL)	10.6 ± 23.1	19.8 ± 22.3	23.8 ± 25.1	0.670
CRP (mg/dL)	21.4 ± 7.5	8.3 ± 7.8	14.3 ± 5.4	**0.003 ***
pH	7.4 ± 0.1	7.4 ± 0.1	7.4 ± 0.1	0.283
Scoring systems				
REMS	5.8 ± 3.4	5.4 ± 3.4	6.43 ± 3.5	0.454
RAPS	2.2 ± 2.1	2.1 ± 2.2	2.4 ± 2.0	0.630
MEWS	3.32 ± 1.8	3.0 ± 1.62	3.86 ± 2.1	0.26
MEDS	8.7 ± 5.0	6.6 ± 4.0	12.4 ± 4.4	**<0.001 ***
NEWS	5.3 ± 3.4	3.9 ± 2.8	8.0 ± 3.0	**0.001 ***
qSOFA	0.6 ± 0.7	0.4 ± 0.5	0.9 ± 0.8	**0.041 ***
Clinical management				
O2 use	22(57.9%)	8(33.3%)	14(100.0%)	**<0.001 ***
Vasopressor use	6(15.8%)	1(4.2%)	5(35.7%)	**0.0018 ***

^a^ Other: Dysuria. * P < 0.05, Statistically significant. Abbreviations: ALK-P, Alkaline phosphatase; ALT, Alanine aminotransferase; AST, Aspartate aminotransferase; BT, body temperature; CKD, chronic kidney disease; CRP, c-reactive protein; Cre, Creatinine; DM, Diabetes Mellitus; GCS, Glasgow coma scale; GI, gastrointestinal; HR, heart rate; Hb, hemoglobin; LDH, Lactic dehydrogenase; LOS, Length of stay; MAP, mean blood pressure; MEDS, Mortality in Emergency Department Sepsis Score; MEWS, Modified Early Warning Score; NEWS, National Early Warning Score; Plt, platelet; qSOFA, quick Sequential Organ Failure Assessment; RAPS, Rapid Acute Physiology Score; REMS, Rapid Emergency Medicine Score; RR, respiratory rate; SBP, systolic blood pressure; WBC, white blood cells.

**Table 2 biology-10-01073-t002:** Results of univariate analyses for predisposing factors on clinical outcomes in patients of *Listeria monocytogenes* bacteremia.

Characteristics	Hazard Ratios	95% Confidence Interval	P Value
Age (years)	1.02	(0.99–1.05)	0.267
Male	0.73	(0.24–5.14)	0.594
Clinical syndromes			
Fever	0.64	(0.21–1.95)	0.43
GI	1.43	(0.46–4.44)	0.531
Respiratory	5.66	(1.57–20.37)	**0.008 ***
Neurology	0.66	(0.19–2.25)	0.508
Other ^a^	0.05	(0.00–219)	0.735
Clinical conditions			
Shock	4.46	(1.39–14.38)	**0.012 ***
Meningitis	0.79	(0.24–2.57)	0.696
Vital signs			
SBP (mmHg)	0.98	(0.94–1.04)	0.186
MAP (mmHg)	0.96	(0.92–1.00)	0.069
HR (bpm)	1.02	(1.00–1.04)	0.119
RR (bpm)	1.34	(1.09–1.64)	**0.005 ***
BT (°C)	1.35	(0.78–2.35)	0.281
SpO2 (%)	0.91	(0.84–0.99)	**0.032 ***
GCS	0.93	(0.81–1.07)	0.303
Comorbidities			
Cardiovascular disease	1.86	(0.62–5.60)	0.269
DM	1.19	(0.26–5.45)	0.825
CKD	2.06	(0.67–6.35)	0.209
Chronic liver disease	2.23	(0.48–10.39)	0.305
Neoplasm	5.00	(1.35–18.53)	**0.016 ***
Autoimmune	0.04	(0.00–12.89)	0.267
Maternal–fetal	0.05	(0.00–114352)	0.682
Immunocompromised	29.17	(0.06–13487)	0.281
Laboratory data			
WBC (counts/uL)	1.00	(1.00–1.00)	0.163
Hb (g/dL)	0.92	(0.74–1.16)	0.485
Plt (×10^3^ counts/uL)	1.00	(0.99–1.00)	0.265
Cre (mg/dL)	1.03	(0.85–1.25)	0.754
ALK-P (U/L)	1.001	(1.0001–1.002)	**0.033 ***
AST (U/L)	1.003	(1.001–1.004)	**0.007 ***
ALT (U/L)	1.01	(1.0004–1.01)	**0.038 ***
LDH (U/L)	1.002	(1.001–1.00)	**0.001 ***
CRP (mg/dL)	1.09	(1.01–1.17)	**0.031 ***
Lactate (mg/dL)	1.02	(0.99–1.04)	0.168
pH	2.33	(0.00–265)	0.814
Scoring systems			
REMS	1.15	(0.96–1.39)	0.128
RAPS	1.05	(0.82–1.34)	0.701
MEWS	1.34	(1.03–1.75)	**0.028 ***
MEDS	1.30	(1.13–1.49)	**<0.001 ***
NEWS	1.41	(1.18–1.68)	**<0.001 ***
qSOFA	2.77	(1.28–5.99)	**0.01 ***
Clinical management			
O2 use	55.21	(0.58–5220)	0.084
Vasopressor use	5.31	(1.66–17.00)	**0.005 ***

^a^ Other: Dysuria. * P < 0.05, Statistically significant. Abbreviations: ALK-P, Alkaline phosphatase; ALT, Alanine aminotransferase; AST, Aspartate aminotransferase; BT, body temperature; CKD, chronic kidney disease; CRP, c-reactive protein; Cre, Creatinine; DM, Diabetes Mellitus; GCS, Glasgow coma scale; GI, gastrointestinal; HR, heart rate; Hb, hemoglobin; LDH, Lactic dehydrogenase; LOS, Length of stay; MAP, mean blood pressure; MEDS, Mortality in Emergency Department Sepsis Score; MEWS, Modified Early Warning Score; NEWS, National Early Warning Score; Plt, platelet; qSOFA, quick Sequential Organ Failure Assessment; RAPS, Rapid Acute Physiology Score; REMS, Rapid Emergency Medicine Score; RR, respiratory rate; SBP, systolic blood pressure; WBC, white blood cells.

**Table 3 biology-10-01073-t003:** Results of univariate and multivariate logistic regression analyses for predisposing factors on the clinical outcomes in patients of *Listeria monocytogenes* bacteremia.

Variables	Univariate			Multivariate		
	HR	95% CI	P Value	HR	95% CI	P Value
CRP	1.09	(1.01–1.17)	0.031 *	1.07	(0.98–1.17)	0.155 ^a^
MEDS	1.30	(1.13–1.49)	<0.001 **	1.25	(1.09–1.43)	0.001 **
NEWS	1.41	(1.18–1.68)	<0.001 **	1.35	(1.11–1.64)	0.003 **

Cox regression. * P < 0.05, ** P < 0.01. ^a^ Multivariate logistic regression analysis adjusted by CRP. CI, Confidence interval; CRP, c-reactive protein; HR, Hazard Ratios; MEDS, Mortality in Emergency Department Sepsis Score; NEWS, National Early Warning Score.

**Table 4 biology-10-01073-t004:** The AUC of ROC, cut-off point, sensitivity specificity, positive predictive value (PPV), negative predictive value (NPV), accuracy, and standard error (SE) of MEDS and NEWS to predict the mortality risk in bacteremic patients of *Listeria monocytogenes*.

Variables	AUC	Cut-Off Point	Sensitivity	Specificity	PPV	NPV	Accuracy	SE	P Value
MEDS	0.829	10	78.6%	79.2%	68.8%	86.4%	78.9%	0.08	0.001 **
NEWS	0.815	8	57.1%	91.7%	80.0%	78.6%	78.9%	0.07	0.001 **

** P < 0.01, Statistically significant; AUC, Area under the curve; MEDS, Mortality in Emergency Department Sepsis Score; NEWS, National Early Warning Score; ROC, Receiver operating characteristic curve.

## Data Availability

Readers could access the data and material supporting the conclusions of the study by contacting Sung-Yuan Hu at song9168@pie.com.tw.

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
