# Peer review of "Performance of Scoring Systems in Predicting Clinical Outcomes in Patients with Bacteremia of Listeria monocytogenes: A 9-Year Hospital-Based Study"

_biology, 2021, doi:10.3390/biology10111073_

Round 1

Reviewer 1 Report

General comments.

The manuscript by Huang et al is revised version of a manuscript that aims to test the performance of various clinical scoring systems for predicting death among patients with Listeria monocytogenes bacteremia. The manuscript has been improved by better editing, although some syntax and wording issues remain, and by a new multivariate analysis that was previously lacking. Some questions about the results of the univariate analysis in table 2 showing significant differences between mortality and survival groups of parameters that were not significantly different in the data presented in table 1 need further explanation.

Specific comments

  • The univariate analysis in table 2 shows significant differences between mortality and survival groups in parameters that were not significantly different in the data presented in table 1. These include Respiratory rate and shock despite the findings that HR, MAP, BP, lactate, WBC, Cre and others were not different. This needs further explanation and suggests liver function (AST, ALT, ALK-P) was over-weighted in the definition of shock.
  • Acronyms of NEWS and MEDS should be written out in the abstract.
  • There are still some editing issues. e.g. lines 31 (“at” should be removed, 46 (“antibiotic” not “antibiotics”), line 67 (“of” should be “for”), 166 (“were” should be ‘are”), Line 212 (what is meant by “the complication rate of meningitis” is not clear), line 235 (“of solid organ” should be ”with solid organ”), 242 (“have should be “had”), line 242 (“Not to ignore is possible” should be “Not to ignore the possibility of”), line 258 (“to” should be deleted), line 298 (“should bear in mind of” should be “should have a”),
  • Lines 246 and 247 – if this was not a food source what are the alternatives?
  • Lines 282-285, the sentence beginning on 282 makes no sense.

Author Response

Dear Reviewer: 

Thanks for your comments to strengthen our manuscript. 

We have point-to-point responses and make the corrections according to the reviewer's comments. 

Please check the attacked PDF file. 

Best regards, 

Sung-Yuan Hu

Reviewer 2 Report

The authors should be commended on their focus and examination of prognostic scoring systems for Listerial bacteremia,  an uncommon presentation but one that is associated with a high mortality rate.  In their case series they identified 39 subjects with Listerial bacteremia and examined several prognostic scoring systems that could predict mortality in these patients.  Results from their study determined that the Mortality in Emergency Department sepsis (MEDS) and National Early Warning Score (NEWS) both had sufficient sensitivity and specificity to identify those patients most at risk for death.  Results from the  study were well presented with adequate tables and graphs to analyze the data.  Although the study found that two scoring systems could prognosticate mortality in patients with Listerial bacteremia, how this information will significantly impact clinical treatment is not clear.  Patients with Listerial infection would likely already be on antibiotics prior to culture results confirming bacteremia and there are no other therapies to add apart from antibiotics.  However, the information would be useful to inform families of the likelihood of death in such cases.  A more important question is if there is a scoring system that can determine which patients with Listerial infection but without initial bacteremia progress to bacteremia and severe disease including septic shock.  Accordingly, more aggressive monitoring and treatment could be utilized in these patients thereby decreasing mortality.  Do the authors have data in their series of patients of the total number with Listerial infections and the number that progressed to bacteremia?  Were there any prognostic indicators that predicted bacteremia in those patients with Listeria as a whole? Such findings would be more clinically relevant in managing these patients and more useful for the clinical community as a whole.   

Author Response

Dear Reviewer:

Thanks for your comments to strengthen our manuscript. 

We have point-to-point responses and make corrections according to the reviewer's comments. 

Please check the attached PDF file. 

Best regards, 

Sung-Yuan Hu

Round 2

Reviewer 2 Report

No additional comments.

Author Response

Dear Reviewers:

Thanks for the reviewers' comments to strengthen our manuscript. The authors revise and edit the manuscript according to the corrections of native English speaker. The authors do point-to-point response according to the reviewer's comments. The point-to-point response is listed in the following table (attached table).

We hope you will appreciate our work. 

Thanks for your attention to our work.

Best regards, 

Sung-Yuan Hu 
